# Solubility and Activation of Hydrogen in the Non-Catalytic Upgrading of Venezuela Orinoco, China Liaohe, and China Fengcheng Atmospheric Residues

**Shunfeng Ji** * and **Anran Zeng**

School of New Materials and Shoes & Clothing Engineering, Liming Vocational University, 298 Tonggang West Street, Quanzhou 362000, China; 20122721@lmu.edu.cn
* Correspondence: 20182807@lmu.edu.cn; Tel.: +86-157-1596-2170

**Abstract:** The solubility of hydrogen in the Venezuela Orinoco, China Liaohe, and China Fengcheng atmospheric residues under reaction conditions of 400 °C, 4 MPa for 20 min was analyzed by determining the composition and structure changes of the products. Activation of hydrogen during the upgrading process was also determined and discussed by the probe method. The results show that lighter components produced in the reaction can increase the hydrogen solubility as the reaction proceeds, and the lighter components present at the liquid level have positive effects on the transfer of hydrogen from the gas phase to the liquid phase. Naphthenic aromatic structures, sulphur and metals have a positive effect on hydrogen activation in the trend of naphthenic aromatic structures > sulphur > metals. Moreover, when sulphur is present, nickel tetraphenylporphyrin has a better effect on hydrogen activation than Vanadium tetraphenylporphyrin. During upgrading, the Venezuela Orinoco atmospheric residue with more sulphur, metals and naphthenic aromatic structures can activate more hydrogen. Both the hydrogen solubility and residue composition have significant effect on the upgrading process.

**Keywords:** hydrogen solubility; hydrogen activation; upgrading; residues



## 1. Introduction

Hydrogen plays an important role in the visbreaking of heavy oils. The solubility of hydrogen in various feed stocks was studied to improve the visbreaking process [1–6]. The results show that the hydrogen solubility increases with increasing temperature and pressure [1–6], while the solubility of hydrogen is lower in oils with more condensed aromatics and hetero-atoms [4]. In order to ensure the accuracy of the data, most studies of hydrogen solubility were performed under the temperatures prior to the reaction point. Changes in the solubility of hydrogen during the reaction have rarely been reported.

In the upgrading of oils under hydrogen, hydrogen gas can function as an extra source of hydrogen radicals to inhibit the cracking and condensation of the oil radicals. The amount of hydrogen radicals has a great impact on the effectiveness of heavy oil upgrading. Investigations [7–10] have shown that the presence of sulphur and metals (such as nickel and vanadium) in heavy oils has a positive effect on hydrogen activation. The hydrogen sulfide formed during the thermal process may produce hydrogen sulfide radicals, which will then react with hydrogen molecules, finally producing hydrogen sulfide again in addition to hydrogen radicals. In the process, the hydrogen sulfide acts as "hydrogen shuttle" to activate the hydrogen [8]. The metals, such as nickel and vanadium, removed from oils partly form corresponding sulfides, which may have catalytic activity [9,10]. There are no quantified data in this aspect.

Based on the previous work [11], hydrogen solubility in residues under reaction conditions were discussed by analyzing the boiling range and structural composition of the

upgrading products. In addition, the probe method was applied to analyse the activation of hydrogen molecules in the upgrading process of residues.

## 2. Experimental Section

### 2.1. Materials

Venezuela Orinoco, China Liaohe, and China Fengcheng atmospheric residues (labeled as VNAR, LHAR and KRAR separately) with initial boiling points of about 350 °C were used as feedstocks in this study. Some of the properties of the feedstocks are shown in Table 1. Hydrogen gas and nitrogen gas both have a purity of 99.999%. Anthracene (labeled as ANT), 9,10-dihydroanthracene (labeled as DHA), sulphur (labeled as S) and metalloporphyrins (nickel tetraphenylporphyrin and vanadium tetraphenylporphyrin, labeled as Ni-TPP and VO-TPP separately) are analytical reagent bought from Sinopharm Chemical Reagent Co., Ltd. which is located in Shanghai, China.

**Table 1.** Properties of the Residues and the Products.

| Sample | VNAR | LHAR | KRAR | P-VNAR | P-LHAR | P-KRAR |
|---|---|---|---|---|---|---|
| Density, g/cm$^3$ (20 °C) | 1.0297 | 0.9821 | 0.9650 | | | |
| dynamic viscosity, mPa·s (100 °C) | 2995.0 | 476.6 | 765.3 | 249.3 | 155.5 | 126.6 |
| molecular weight | 781.7 | 686.3 | 736.8 | 586.6 | 620.7 | 702.9 |
| elemental analysis | | | | | | |
| C, wt % | 83.82 | 86.83 | 85.89 | 84.28 | 86.86 | 86.55 |
| H, wt % | 9.96 | 11.42 | 11.53 | 10.18 | 11.49 | 12.03 |
| S, wt % | 4.31 | 0.36 | 0.36 | 3.76 | 0.38 | 0.23 |
| N, wt % | 0.71 | 0.86 | 0.92 | 0.81 | 0.97 | 0.83 |
| H/C atomic ratio | 1.416 | 1.567 | 1.600 | 1.439 | 1.576 | 1.656 |
| Nickel, ppmw | 113 | 68.7 | 43.5 | | | |
| Vanadium, ppmw | 417 | 1.81 | 0.77 | | | |
| average structural parameters [a] | | | | | | |
| Aromaticity, % | 0.333 | 0.247 | 0.216 | 0.327 | 0.243 | 0.190 |
| number of paraffinic rings | 3.509 | 2.726 | 3.194 | 2.714 | 2.551 | 2.642 |
| number of aromatic rings | 4.039 | 2.568 | 2.341 | 2.864 | 2.227 | 1.911 |
| boiling range, °C | | | | | | |
| IBP~350 | 1.5 | 1.8 | 1.5 | 10.7 | 5.0 | 6.0 |
| 350~420 | 3.5 | 5.2 | 10.1 | 18.3 | 9.5 | 11.7 |
| 420~500 | 19.0 | 24.0 | 20.9 | 26.0 | 27.0 | 22.8 |
| 500~FBP | 76.0 | 69.0 | 67.5 | 45.0 | 58.5 | 59.5 |

[a] The average structural parameters were calculated by the modified Brown–Ladner (B–L) method, based on $^1$H-NMR.

### 2.2. Upgrading Processing

The upgrading of the residues was carried out in a 300 mL high-pressure autoclave fitted with a two-blade magnetically driven agitator. About 60 g of residue was put into the autoclave. The autoclave was then purged and charged with the gas (hydrogen or nitrogen). A total of 4 MPa of initial pressure was charged at room temperature. Then the autoclave was heated to a reaction temperature of 400 °C and held for 20 min. Throughout the process, the speed of the agitator was 300 rpm. The autoclave was cooled by cold water immediately after the reaction was completed. Some of the properties of the products were then characterized.

### 2.3. Determination of Hydrogen Transfer Ability

Hydrogen donating ability (HDA) and hydrogen accepting ability (HAA) are generally used as parameters to determine the transferable hydrogen in the oil systems. ANT was used as a hydrogen accepting probe to determine the HDAs of the residues, both at the atmosphere of nitrogen and hydrogen; DHA was chosen as a hydrogen donating probe to measure the HAA of the residues under nitrogen [8]. A total of 0.5 g residue and 0.5 g probe were put into a 30 mL micro-autoclave. The micro-autoclave was then purged and charged with the gas (hydrogen or nitrogen) to an initial pressure of 4 MPa at room temperature.

After being reheated in tin bath at 350 °C for 10 min, the micro-autoclave was transferred into another tin bath at a reaction temperature of 400 °C and held for 20 min. After the reaction was completed, the micro-autoclave was cooled by cold water. Then the products were flushed out completely by toluene and characterized by gas chromatography [8]. The HDA and HAA of the residues were calculated according to the peak areas of anthracene, 9,10-dihydroanthracene and tetrahydracene in the spectrogram. Both HDA and HAA were expressed as milligrams of hydrogen per gram of residue (mg H/g oil). Each test was conducted three times, and typical errors of calculations are within 0.001 mg H/g oil.

### 2.4. Preparation of Metalloporphyrin-Anthracene Mixtures

In order to ensure the uniformity of the mixtures, metalloporphyrin and anthracene should be mixed beforehand. A total of 10 mg of metalloporphyrin and 5 g of anthracene were dissolved in toluene and dichloromethane separately, and then mixed. The mixture was heated to reflux for 30 min at 120 °C. Finally, the mixture evaporated most of the solvent and was vacuum dried.

## 3. Results and Discussion

### 3.1. Hydrogen Solubility in Reaction

In our previous work [4], hydrogen had a higher solubility in KRAR than LHAR and VNAR at the same conditions; VNAR has the lowest hydrogen solubility in the three residues. The determination was conducted prior to the reaction point. While under the reaction conditions, hydrogen solubility was changing all the time with the changing of residue composition and structures, although the temperature and pressure were the same. Then the three residues were upgraded under hydrogen at 400 °C, 4 MPa for 20 min. The boiling range and average structural parameters of the products (labeled as P-VNAR, P-LHAR and P-KRAR separately) were characterized by gas chromatography and $^1$H-NMR. As shown in Table 1, percentages of boiling ranges IBP-420 °C for the three products were 29.0% (P-VNAR), 14.5% (P-LHAR) and 17.7% (P-KRAR), which are markedly different from the initial residues of 5.0% (VNAR), 7.0% (LHAR) and 11.6% (KRAR) separately, especially for VNAR. Moreover, the $^1$H-NMR analysis shows that the number of aromatic rings of VNAR, LHAR and KRAR decreases 1.175, 0.341 and 0.430 separately; VNAR has a higher cracking degree. Much lighter components are produced in VNAR. These lighter components mainly originated from breaking of alkyl side chains and cracking of big molecules with condensed rings. Additionally, hydrogen solubility in paraffins increases as the length of chains increases; the hydrogen solubility in aromatic hydrocarbons decreases as the numbers of aromatic rings increase [12,13]. Furthermore, alkanes have higher hydrogen solubility than aromatics [12,13]; thus, a higher content of lighter components implies a higher hydrogen solubility. Moreover, the amount of lighter components that exist on the liquid level have positive effects on hydrogen molecules transferring from a gas phase to a liquid phase. Thus, as the reaction proceeds, there is likely to be a greater increase in hydrogen solubility in VNAR than in LHAR and KRAR.

Stability of the three products was determined by spot test [14]. The upgrading products of the three residues were put under dark conditions for 45 days, and then their stability was tested according to the spot test steps. The stability was judge by the spot test evaluation criterion [14]. As shown in Figure 1, the spot of P-VNAR is more homogeneous than the other two products. The spots of P-LHAR and P-KRAR obviously have out-ring, especially for P-KRAR. Thus, the stability exhibits the trend of P-VNAR > P-LHAR > P-KRAR. The upgrading effect of P-VNAR is the best. This result may be due to the deceasing hydrogen solubility and the better hydrogen activation in VNAR in the reaction process.

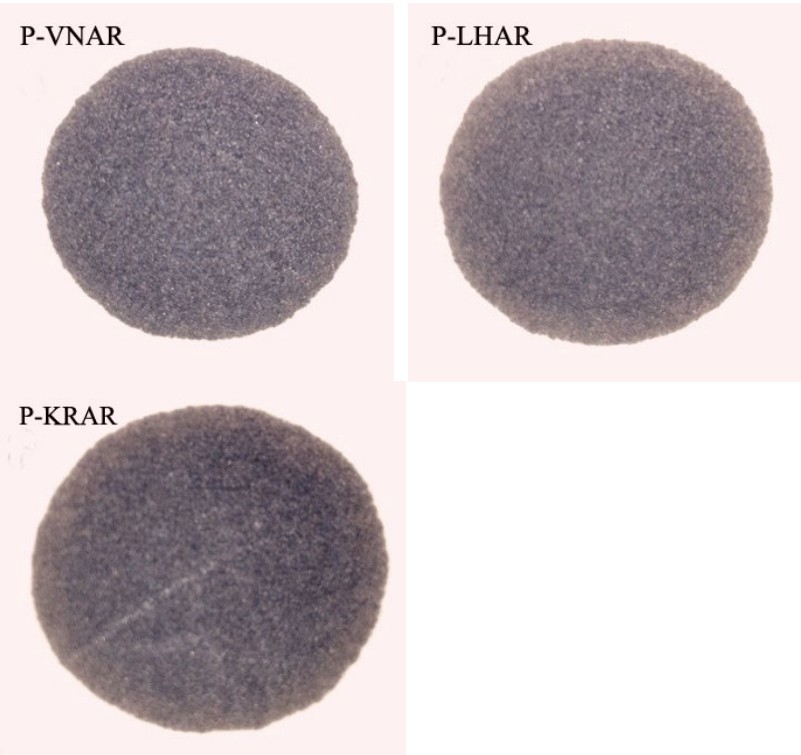

**Figure 1.** Spot test of the products.

### 3.2. Hydrogen Transfer of the Residues

Hydrogen transfer of the three residues was analyzed by measuring their HDA and HAA at 400 °C at 4 MPa for 20 min. DHA was used as a hydrogen donating probe to determine the HAA of the three residues under nitrogen, and ANT was used as a hydrogen accepting probe to determine the HDA under nitrogen and hydrogen, respectively. The results are shown in Figures 2 and 3.

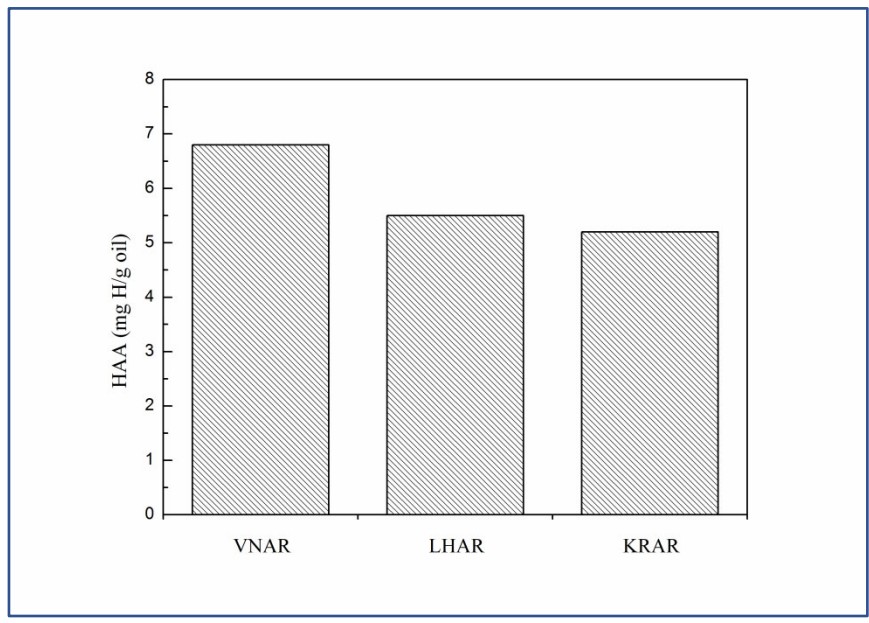

**Figure 2.** HAA of the three residues.

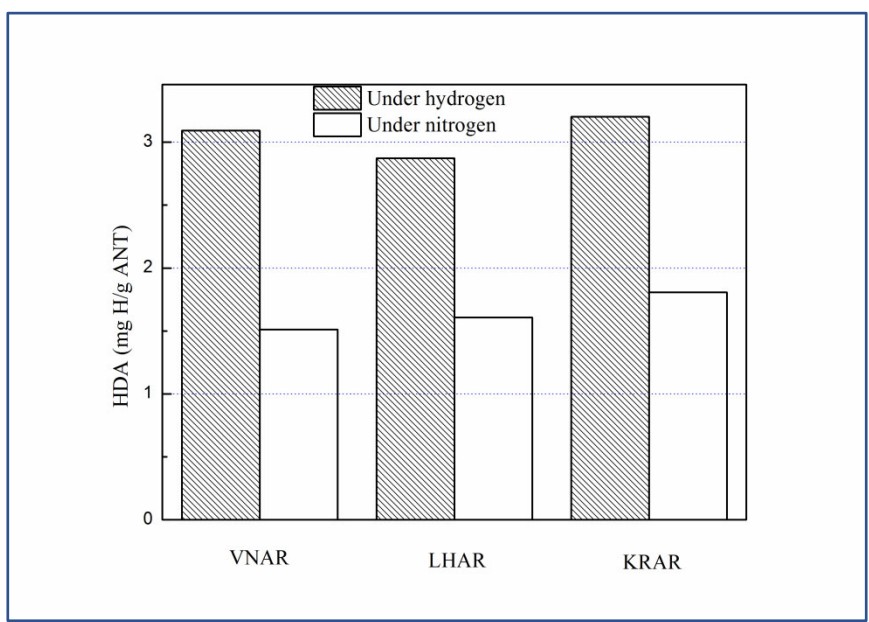

**Figure 3.** HDA of the three residues under hydrogen and nitrogen.

As shown in Figure 2, the VNAR with the highest viscosity has a much higher HAA, while the KRAR has the lowest HAA of all. On average, the number of molecular and aromatic rings in VNAR is much higher than LHAR and KRAR; it has been indicated that the system of VNAR can accept more hydrogen from outside at the same condition. This property may have an impact on the activation of hydrogen molecules to some extent.

In Figure 3, the HDA of the three residues under nitrogen presents an increasing trend of VNAR < LHAR < KRAR; the trend is completely opposite to their HAA. Higher HAA means lower HDA. However, the trend of HDA is different with the presence of hydrogen. The HDA of the VNAR system obviously increases, and is higher than LHAR, but lower than KRAR. By calculating the differences between HDA under hydrogen and under nitrogen, we can see that the difference values of VNAR, LHAR and KRAR are 1.581 mg H/g oil, 1.264 mg H/g oil and 1.395 mg H/g oil, respectively. The value of VNAR is much higher than the other two residues, which indicates that, minus the effect of the residue itself, hydrogen gas donates more hydrogen in the VNAR system. In other words, more hydrogen molecules are activated in the VNAR system. For LHAR and KRAR, the KRAR system can activate more hydrogen.

### 3.3. Activation of the Hydrogen

In non-catalytic conditions, the activation of hydrogen molecules is only related to the composition of the residue when the temperature and pressure are the same. Naphthenic aromatic structures were shown to play an important role in the activation of hydrogen molecules by acting as "hydrogen shuttle". These structures can be hydrogenated and then donate the hydrogen out of which the process promotes the activation of hydrogen molecules. Thus, residues containing more naphthenic aromatic structures can activate more hydrogen molecules. As shown in Table 1, the number of paraffinic rings of the three residues is in the trend of VNAR > KRAR > LHAR. The number of paraffinic rings is directly related to the amount of naphthenic aromatic structures. Thus, VNAR and KRAR have greater advantages in terms of hydrogen transfer or hydrogen activation. However, the number of aromatic rings in VNAR is 4.039, which is even twice as much as the other residues. This suggests that VNAR needs more hydrogen to inhibit its own condensation. Theoretically, the HDA of KRAR should be the highest; but in fact it is the HDA of VNAR that is the highest. Therefore, the naphthenic aromatic structure is not the only factor affecting hydrogen activation.

As mentioned earlier, S and metals existing in residues may have a positive effect on the activation of hydrogen; the hydrogen sulfide formed during thermal process may transfer the hydrogen by "hydrogen shuttle" [8]. In addition, metals (such as Ni and V) removed from oils partly form corresponding sulfides, which may have catalytic activity. Some reactions were designed to prove and quantify the impact of S and metals (which existed in the form of metalloporphyrin). The reactions are shown as below.

① $H_2$ + ANT
② $H_2$ + S +ANT
③ $H_2$ + metalloporphyrin + ANT
④ $H_2$ + S +metalloporphyrin + ANT

Reaction system ① was to determine the activation ability of hydrogen itself by determining the HDA of hydrogen. Reaction system ② ③ ④ were used to analyze the effects of S and metalloporphyrin on the activation of hydrogen, and also the function between S and metalloporphyrin. The reactions were conducted at an initial pressure of 3 MPa at temperature points of 370, 390, 410 and 430 °C for 10 min. The results are shown in Figures 4 and 5.

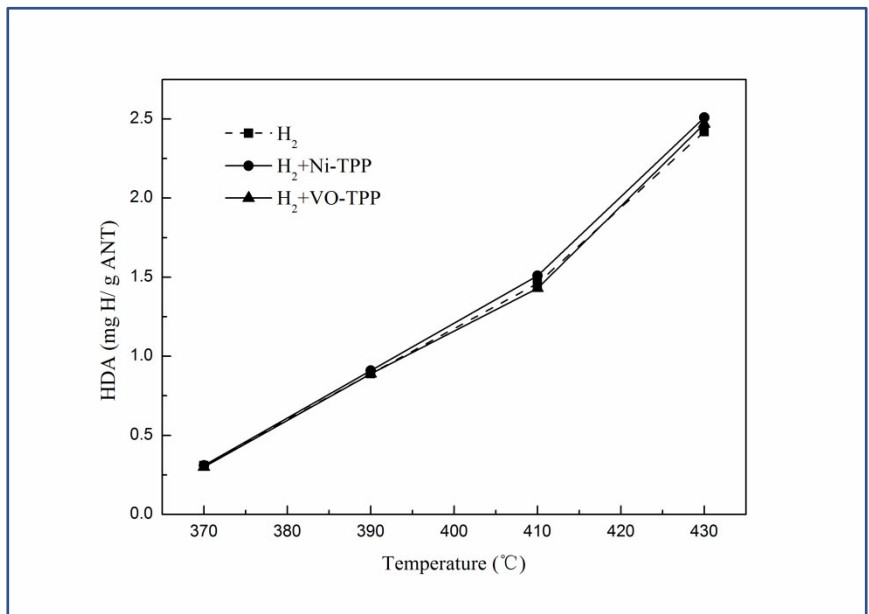

**Figure 4.** HDA of hydrogen with Ni-TPP and VO-TPP.

As shown in Figure 4, the activation capacity of hydrogen itself was increasing as the temperature increased. After adding Ni-TPP and VO-TPP, respectively, the HDA of the system was almost unchanged. This phenomenon indicates that the metalloporphyrin has no catalysis under the reaction conditions.

Then the hydrogen system added some S of 5% in quality, imitating the content in real residues. The results in Figure 5 show that the HDA of the system increased significantly with the addition of S, which indicates that S promotes the activation of hydrogen, and the effect is prominent. Then Ni-TPP and VO-TPP were added in system ②, respectively; results were also shown in Figure 5. It is indicated that both Ni-TPP and VO-TPP have a positive impact on the HDA of the hydrogen-S system, the HDA has different degrees of improvement. The impact of Ni-TPP is better than VO-TPP. Some demetalization products with catalytic activity were produced in the system, and the catalytic activity caused by Ni-TPP was better than VO-TPP.

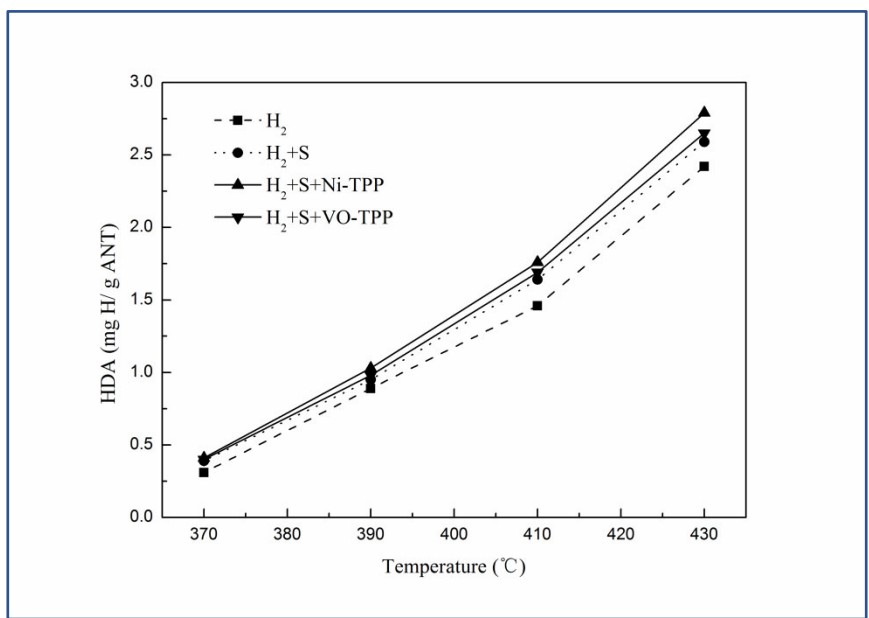

**Figure 5.** HDA of hydrogen with S and metalloporphyrin.

For all three residues, the contents of S and metals in VNAR are much higher than LHAR and KRAR (as shown in Table 1). Thus, VNAR has greater advantages for the activation of hydrogen. However, the reaction of demetalization is difficult to achieve in the real residue system [15], while the desulphurization reaction is much easier to conduct. Therefore, in real reaction systems, the effect of S is effective and prominent, while the function of metals is limited.

In conclusion, considering the content of the naphthenic aromatic structures, S and metals in the three residues, and the difficulty of each reaction, the factors influencing hydrogen activation have a positive functions in the trend of naphthenic aromatic structures > S > metals. VNAR containing all the advantages can activate more hydrogen and has the better upgrading effect.

## 4. Conclusions

Hydrogen solubility in residues under reaction conditions is much different from that under non-reactive conditions. Lighter components produced in the reaction will increase hydrogen solubility with the reaction proceeds, and the lighter components existing on the liquid level have positive effects on hydrogen molecules transferring from gas phase to liquid phase. Naphthenic aromatic structures, S and metals in the oils have a positive impact on hydrogen activation, and the positive functions are in the trend of naphthenic aromatic structures > S > metals. In the case of metals, Ni-TPP has a better effect on hydrogen activation than VO-TPP when S exists. VNAR with more S, metals and naphthenic aromatic structures can activate more hydrogen. Both hydrogen solubility and residue composition have big effects on upgrading effect.

**Author Contributions:** Conceptualization, S.J. and A.Z.; methodology, S.J.; software, S.J.; validation, S.J. and A.Z.; formal analysis, S.J.; investigation, S.J.; resources, S.J.; data curation, S.J.; writing—original draft preparation, S.J.; writing—review and editing, A.Z.; visualization, S.J.; supervision, S.J.; project administration, S.J.; funding acquisition, S.J. All authors have read and agreed to the published version of the manuscript.

**Funding:** Ph.D. Research Foundation of Liming Vocational University (LZB201901).

**Conflicts of Interest:** The authors declare no competing financial interest.

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
