# Peer review of "Solubility and Activation of Hydrogen in the Non-Catalytic Upgrading of Venezuela Orinoco, China Liaohe, and China Fengcheng Atmospheric Residues"

_processes, doi:10.3390/pr9122274_

Round 1
Reviewer 1 Report
The manuscript needs to be thoroughly revised especially in terms of grammatical errors. Also, the objective needs to be clearly stated and the introduction section needs to be significantly improved. Furthermore, the title mentioned 'residue' without providing where the residue is coming from. The title needs to be revised as well

Reviewer 2 Report
The work is interesting and relevant. However, there are some comments and recommendations.
- Lines 15, 16 - please decipher the abbreviation
- In the Introduction part, it is necessary to expand the literature review related to the hydrogen shuttle
- Please describe all methods in more detail, for example HDA and HAA.
- Perhaps the paragraph line 87-99 needs to be moved to the Introduction part
- Figure 1. How was stability determined? Visually? It is necessary to indicate somewhere. What can be seen in Figure 1? - describe in more detail. Unfortunately, in this form it is not clear
- It would be desirable to bring the chemical reactions themselves, if possible. How the presence of naphthenic aromatic structures, sulfur and metal porphyrins contributes to the activation of hydrogen
The article can be published after correcting the comments.
Round 2
Reviewer 1 Report
The reviewer appreciates the authors for revising the manuscript based on the comments and suggestions provided by the reviewer. This has greatly improved the manuscript and it is in good shape for publication